# Reduction of Volatile Organic Compounds (VOCs) Emissions from Laundry Dry-Cleaning by an Integrated Treatment Process of Condensation and Adsorption

**Mugeun Song †, Kyunghoon Kim †, Changmin Cho and Daekeun Kim \***

Department of Environmental Engineering, Seoul National University of Science and Technology, Seoul 01811, Korea; qmfel14@naver.com (M.S.); kyunghoon.kim@seoultech.ac.kr (K.K.); ckdals6284@naver.com (C.C.)

\*   Correspondence: kimd@seoultech.ac.kr; Tel.: +82-2-970-6606

†   These authors contributed equally to this work.

**Abstract:** Volatile organic compounds (VOCs) are intermittently emitted at high concentrations (tens of thousands of ppmv) from small-scale laundry shops in urban areas, affecting the urban atmospheric environment. In this study, we suggested integrating VOC treatment processes incorporating condensation and adsorption in series to remove VOCs released from small-scale laundry dryers (laundry weighing less than 30 kg). We designed two different processes depending on regeneration modes for adsorber beds; an open-circuit flow process and a closed-loop flow process in regeneration mode. Our VOC treatment processes enable sustainable operation via the regeneration of adsorbers on a regular basis. Before applying the VOC treatment processes, average concentration of total volatile organic compounds (TVOCs) was 4099 ppmv (12,000 ppmv of the peak concentration) during the drying operation. After applying our closed-loop flow process, TVOC concentration decreased to 58 ppmv, leading to 98.5% removal efficiency. We also verified the robustness of our process performance in a continuous operation (30 cycles) by using a process simulation program. Lastly, we observed that our integrated treatment process can contribute to reductions in ozone and secondary organic aerosol generation by 90.4% and 95.9%, respectively. We concluded that our integrated VOC treatment processes are applicable to small-scale laundry shops releasing high-concentration VOCs intermittently, and are beneficial to the atmospheric environment.

**Keywords:** VOC reduction; laundry dry-cleaning; condensation; adsorption; regeneration

## 1. Introduction

Volatile organic compounds (VOCs) originate from natural and/or anthropogenic emission sources, affecting atmospheric environment [1–3]. The anthropogenic emission sources of VOCs include internal combustion engine vehicles, large-scale industrial complexes, residential combustion, and solvent use [4–6]. Once VOCs are released from their original sources, they have adverse effects on tropospheric chemistry. To be specific, VOCs with high reactivity are known to behave as major precursors to produce ozone ($O_3$) and secondary organic aerosols (SOA) via photochemical chemical reactions [7–9]. Moreover, parts of the VOCs have adverse health effects [10,11] and thus were included in the 187 toxic air pollutants designated by the United States Environmental Protection Agency [12].

Among the various anthropogenic emission sources of VOCs, half of VOC emissions are produced by the use of volatile chemical products in industrialized cities in the United States, while the contribution of vehicle emissions has decreased at the same time [5]. McDonald et al. also found that VOC emission factors (defined as the emitted amount of VOC per unit product use) of volatile chemical products are one to two orders of magnitude higher than those of transportation (vehicle emissions). As people in developed countries spend more than 80% of their time indoors [13], human exposure to VOCs in indoor environments,

primarily caused by the indoor use of volatile chemical products, can receive increased public attention. Volatile chemical products mainly include pesticides, inks, adhesives, coatings, household cleaning agents, detergents, and personal care products [5,14].

Laundry facilities are equipped in the laundry shops and most households, and they use detergents that are composed of various VOCs. In general, laundry processes consist of washing to remove stains and drying to remove water and residual detergents. Our previous study found that 36 VOCs were detected from the exhaust gas of laundry drying machines, and the mass of these VOCs emitted from one cycle of drying operation for 3 kg of cotton fiber was approximately 9 g [15]. It was also found that workers in laundry shops in Republic of Korea might have an increased risk of liver malfunctions [16]. Furthermore, compared to other VOC emission sources (e.g., wastewater treatment, industry), laundry facilities are close to urban residential areas, where $NO_x$ and $SO_x$ ambient concentrations are higher than in suburban areas [17,18], which may lead to larger production of ozone and SOA with the same amount of VOCs in urban areas. Thus, efforts to reduce VOC emission during laundry processes are necessary to protect indoor air quality, to reduce the health risks of laundry workers; and outdoor air quality, to decrease the production of ozone and SOA in urban areas.

Reduction processes of VOCs in the air phase include condensation [19,20] and adsorption [21,22]. Condensation has been widely used for recovering VOCs because it requires simple equipment to remove a wide range of VOCs in the air phase and operates at low temperature with lower risk of fire and higher level of operation safety [23]. The condensation process is also useful to remove VOCs in the air phase at high composition ($\geq 1\%$) [24]. However, the operating cost of condensation can be increased for VOCs at relatively low concentrations because the condensation process for low-concentration VOCs should be operated in cryogenic conditions. Adsorption is another popular method for gas-phase VOC reduction because of its various advantages, including selection of adsorbents according to target VOCs and reusability of adsorbents via regeneration [22,25]. Despite these advantages, adsorption is not useful for removing high-concentration VOCs because of the increased time and cost for regenerating adsorbers or replacing them with new ones [23], which may make adsorption unsuitable for the laundry-drying process, which emits high-concentration VOCs intermittently. Therefore, further studies are needed to develop a VOC-reduction process integrating condensation and adsorption into a new configuration, useful for laundry drying.

The objectives of the current study were to develop integrated VOC-reduction processes consisting of adsorption and condensation for small-scale laundry dryers (less than 30 kg of laundry) and evaluate the performance of our processes in multiple ways. Specifically, our integrated VOC-reduction processes were carried out by low-temperature condensation and adsorption/desorption in continuous operation. We prepared two different regeneration modes for adsorber beds; open-circuit flow mode and closed-loop flow mode. We also conducted the process simulation to ensure the performance of our integrated VOC-reduction process in a long-term operation. Lastly, to evaluate our VOC-reduction process designed for small-scale dryers, we assessed the reduction in total volatile organic compound (TVOCs) concentration and the contribution of ozone and SOA production after the VOC-reduction process.

## 2. Materials and Methods

### 2.1. Dry-Cleaning Process

To simulate dry-cleaning process in laundry shops, we set up the actual size of the washer and dryer in our laboratory and operated the dry-cleaning process according to our survey results of laundry shops (i.e., amount of water used in the cleaning process, amount of detergent, drying temperature, drying time) [26]. For the simulation of the dry-cleaning process, we used two separate processes; a cleaning process followed by a drying process using hot air vented off to the atmosphere. We used a 13 kg washing machine (ESE-7313, Eunsung Engineering, Daegu, Korea), a 15 kg dryer (SR-7615, Ensung Engineering, Daegu,

Korea), and an 8 kg/hour electric steam boiler (PHE-5, Pyeonghwa-boiler, Daegu, Korea) to heat the drying air. As a detergent for the cleaning process, we used a petroleum-based organic solvent (KS M 2611DML) that was found to be the most widely used in our previous survey [26]. Primary ingredients of the solvents include n-undecane (43%), decane (34%), n-dodecane (9%), nonane (8%), octane (1%), ethylbenzene (0.87%), meta and para-xylene (0.84%), styrene (0.71%), isopentane (0.66%), and ortho-xylene (0.56%) [15]. For the cleaning and drying processes, we used 3 kg of 100% cotton fiber as laundry. The cleaning process lasted for 23 min and the drying process was operated at 40 °C for 40 min using hot air supplied from the electric steam boiler. More details on the cleaning and drying processes are available elsewhere [15].

### 2.2. VOC-Reduction Process

Conceptual layouts of our integrated treatment processes for VOC reduction are shown in Figure S1. The integrated treatment processes included condensation and adsorption in series. We designed two different processes depending on methods of adsorption bed regeneration: (1) a process with open-circuit flow in regeneration mode and (2) a process with closed-loop flow in regeneration mode. For the open-circuit flow process (Figure S1a), we prepared two individual sets of condensation/adsorption (condenser A → adsorption bed A → adsorption bed B → condenser B, or vice versa). The open-circuit flow process with two individual sets aimed to allow each adsorption bed to adsorb or desorb in turn by reversing the process flow. For the closed-loop flow process (Figure S1b), we prepared one condensation/adsorption set only (condenser → adsorption bed). During the regeneration mode of the adsorption bed in the open-circuit flow process, the condenser reduces the partial pressure of VOCs and then the downstream flow from the condenser desorbs VOCs from the adsorption bed for the regeneration. Both the open-circuit flow process and closed-loop flow process have the two-step cycle including the feeding phase (adsorption) and the purging phase (regeneration).

To be more specific, Figure 1 shows the full-scale schematic layouts of two integrated treatment processes. Firstly, a process with open-circuit flow in regeneration mode (Figure 1a) recovers VOCs existing in the 30–50 °C exhaust gas of a laundry dryer using a condenser. Then, uncondensed VOCs are further introduced into an adsorption bed with a temperature below 0 °C for better adsorption. For the desorption process, feed gas at a temperature of 40 °C is introduced from the upstream condenser/adsorber set to the adsorber of downstream set. Secondly, a process with closed-loop flow in regeneration mode (Figure 1b) enables the internal circulation (2 s per one circulation) of exhaust gas with one set of condenser/adsorber only. The closed-loop flow process aimed to reduce the area needed for the process (area for open-circuit flow process: 1.26 m$^3$, area for closed-loop flow process: 0.66 m$^3$) and further increase the efficiency of VOC reduction from exhaust gas. We regarded 2 adsorptions and 1 desorption as one daily operating cycle with consideration for the daily average number of dry-cleaning operations of laundry shops [26]. Detailed design parameters for the open-circuit flow process and the closed-loop flow process are provided in Table 1.

The condenser employed in our processes used chlorodifluoromethane (CHClF$_2$) as a refrigerant in the indirect evaporative cooling system [27]. We estimated theoretical condensation temperature of total volatile organic compounds (TVOCs) using the Antoine equation (National Institute of Standards and Technology, http://webbook.nist.gov, accessed 25 August 2020) and further tuned the condensation temperature considering the TVOC concentration obtained via our measurement.

The adsorption process was designed to fill adsorbents inside a cylindrical reactor made of aluminum. We prepared the adsorbent by mixing one commercial adsorbent (i.e., activated alumina) [28] and two other adsorbents (i.e., porous clay heterostructures, metal organic frameworks) produced in our laboratory [29,30]. More details on the adsorbents are presented in Table 1.

(a) A process with open−circuit flow in regeneration mode

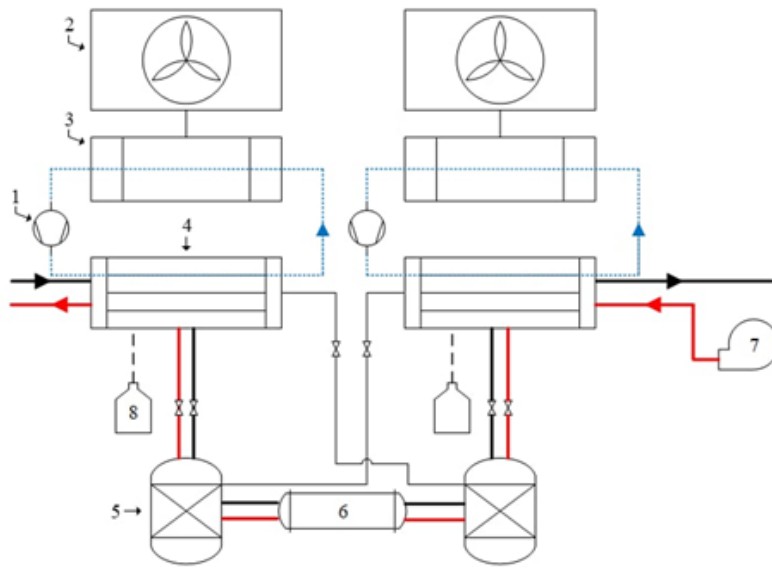

(b) A process with closed−loop flow in regeneration mode

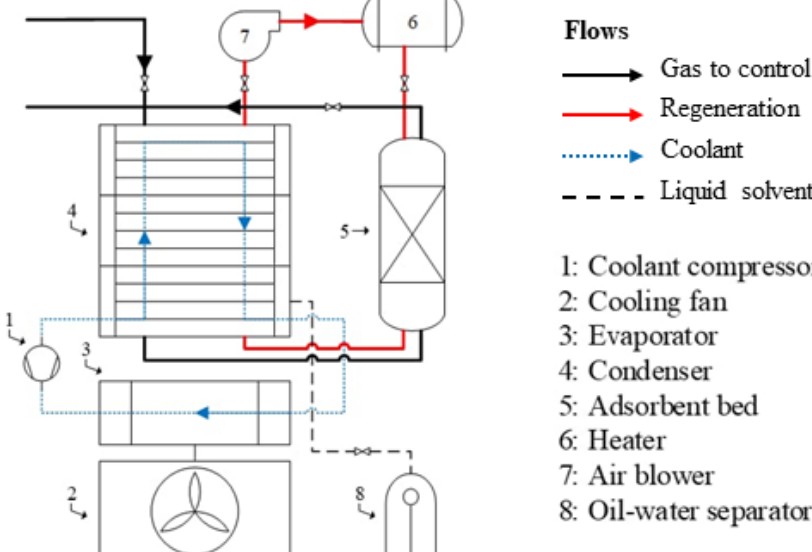

**Flows**

——————▸ Gas to control
——————▸ Regeneration
·············▸ Coolant
– – – – – Liquid solvent

1: Coolant compressor
2: Cooling fan
3: Evaporator
4: Condenser
5: Adsorbent bed
6: Heater
7: Air blower
8: Oil-water separator

**Figure 1.** Schematic layouts of two different integrated treatment processes employing different gas flows and regeneration scheme: (**a**) open-circuit flow in regeneration mode and (**b**) closed-loop flow in regeneration mode. System configuration of both integrated treatment processes enables sustainable operation via regeneration of adsorbent beds.

*2.3. VOCs Sampling and Analysis*

To investigate the VOC-reduction efficiency of our processes, we sampled the exhaust gas before and after the operation of the VOC-reduction processes. We measured the gas flow rate and its temperature using a flowmeter (TSI-9535, TSI, Shoreview, MN, USA). We also measured the TVOC concentration in the exhaust gas flow with a photoionization gas detector (NEO MP 182, mPower Electronics, Santa Clara, CA, USA) that produces isobutylene-based TVOCs concentration with the analytical concentration range of 0.01–15,000 ppmv. Then, we converted the isobutylene-based TVOCs concentration into toluene-based TVOCs concentration using a calibration curve of toluene standard gas.

**Table 1.** Design parameters of the two integrated treatment systems suggested in this study.

| | Open-Circuit Flow Process (1st Prototype) | Closed-Loop Flow Process (2nd Prototype) |
|---|---|---|
| Total system volume | 1.26 m$^3$ (850 × 1100 × 1350 mm) | 0.66 m$^3$ (500 × 800 × 1650 mm) |
| **Condenser** | | |
| Maximum power | 3 kW | 1.5 kW |
| Number of columns | 1 | 3 |
| Condensation temperature | 2 °C (condensation/adsorption) −8 °C (desorption/condensation) | −10 °C (condensation/adsorption) −10 °C (desorption/condensation) |
| **Adsorber** | | |
| Packed volume | 0.008 m$^3$ | 0.008 m$^3$ |
| Adsorbent (packing volume composition) | Activated alumina [a] (95.8) PCH [b] (2.3) MOF [c] (1.9) | Activated alumina [a] (88.9) MOF [c] (11.1) |
| Space velocity | 8694 h$^{-1}$ | 10,928 h$^{-1}$ |
| **Heat exchanger** | | |
| Maximum power | 2 kW | 1 kW |

[a] Activated alumina (M&E Tech, Hwasung, Korea): 0.4 g of dry-cleaning solvent/g of adsorption capacity and 0.64 g/cm$^3$ of density.
[b] Porous clay heterostructures (fabricated in lab.): 0.123 g of toluene/g of adsorption capacity and 0.55 g/cm$^3$ of density. [c] Metal organic frameworks (fabricated in lab.): 0.467 g of toluene/g of adsorption capacity and 0.17 g/cm$^3$ of density.

### 2.4. Process Simulation

Among two integrated treatment processes, we conducted the process simulation with the closed-loop flow process to check the robustness in the performance of the adsorption process with a continuous operation. For the simulation, we used a pore and surface diffusion model (PSDM), which is embedded in an adsorption design software (AdDesignS[TM], Michigan Technological University, Houghton, MI, USA) [31]. Some input parameters for the process simulation were based on a previous adsorption isothermal study [32] and details on the input parameters are available in Supplementary Materials (Table S1). Among the VOCs included in the exhaust gas of laundry dryers, we selected decane ($C_{10}H_{22}$) as a representative compound for the process simulation because decane was known to generate the highest mass from one cycle of drying operation among detergent ingredients [15].

### 2.5. Effect of the VOC-Reduction Process on the Atmosphere

To quantitatively examine the effect of our VOC-reduction process on the atmospheric environment, we calculated photochemical ozone creation potential (POCP) and secondary organic aerosol potential (SOAP) before and after operating our closed-loop flow process. The POCP is a compound-specific value representing the relative ozone formation potential of a specific compound, with the POCP value of ethylene set as 100 (a reference value) [33]. In addition, the SOAP represents the potential of a compound to produce secondary organic aerosols, with the SOAP value of toluene set as 100 (a reference value) [34]. In this study, we estimated the POCP-weighted emissions and the SOAP-weighted emissions (g/dry-cleaning cycle) of each VOC with consideration for the emission amount of individual VOCs [34]. The POCP-weighted emission and SOAP-weighted emission for species, $i$, were calculated using below equations.

$$POCP_{weighted\ emission} = E_i \times POCP_{species,i}$$

$$SOAP_{weighted\ emission} = E_i \times SOAP_{species,i}$$

where $E_i$ represents the emission (g/dry-cleaning cycle) of VOC species $i$, from a one-time dry-cleaning cycle. We obtained the POCP and SOAP values of each compound via literature searches [33,34].

To measure the concentrations of individual compounds, we used Tenax-TA (APK Sorbent Tube, KNR Co., Ltd., Namyangju, Korea) as an adsorbent in adsorbent tubes and analyzed each compound with gas chromatography coupled with mass spectrometry (GC-MS). More details on VOC sampling and analysis are available in our previous study [15].

## 3. Results and Discussion

### 3.1. Determination of VOC-Reduction Process Parameters

Firstly, we conducted condensation experiments with different condensation temperatures to determine the optimal temperature for VOC removal in our condenser (Figure S2). We found the lower condensation temperature led to higher VOC-removal efficiency, and our experimental results fitted with the theoretical removal efficiency obtained from the Antoine equation (Figure S2a). We also examined the effect of both condensation temperature and inlet TVOC concentrations to removal efficiency and found that higher TVOC concentration and lower condensation temperature resulted in higher VOC-removal efficiency in the condenser (maximum 75.1% with 12,000 ppmv and $-8\,°C$) (Figure S2b). Considering these results, we set the optimal condensation temperature of the closed-loop process flow as $-10\,°C$ with three condensation columns inside the condenser (Table 1).

Secondly, we conducted absorption experiments to examine the effects of space velocity and inlet TVOC concentrations on the VOC adsorption capacity of our activated alumina-based adsorber (Figure S3). With toluene as a representative compound, we found that lower space velocity and higher inlet TVOCs concentration led to higher adsorption capacity in the adsorber (maximum 31.7 mg/g with 1800 ppmv and 7000 $h^{-1}$) (Figure S3a). We also conducted adsorption/desorption experiments to obtain the optimal desorption temperature (Figure S3b). We found that 40 °C of desorption temperature resulted in 80.7% desorption of adsorbed toluene. Lastly, we observed that operating temperature (25 °C, 40 °C) did not affect the amount of toluene adsorption, but affected desorption.

### 3.2. VOC Reduction

Figure 2 shows TVOC concentration in exhaust gas emitted from a laundry dryer before and after applying the integrated treatment processes. TVOC concentration before removal of VOCs started from 7000 ppmv, reached 12,000 ppmv as a maximum concentration after approximately 8 min, and then decreased monotonically down to 300 ppmv at the end of the drying operation (40 min). Average TVOC concentration during the drying operation was 4098.8 ± 3598.3 ppmv before applying the VOC-removal processes, and the total emission amount of TVOCs was estimated as 829.0 ± 15.7 g/cycle. After we removed VOCs using our open-circuit flow process, the average TVOC concentration was 140.8 ± 39.0 ppmv and total emission amount was 16.5 ± 5.7 g/cycle, leading to TVOC-removal efficiency of 98.0 ± 0.7%. When we applied the closed-loop flow process, the average TVOC concentration was 57.5 ± 10.6 ppmv, resulting in TVOC-removal efficiency of 98.5 ± 0.3%. Considering the above results, we found that the closed-loop flow process showed the better performance for the VOC removal than the open-circuit flow process. The higher TVOC-removal efficiency of the closed-loop flow process might be caused by the higher number of columns in the condenser (three columns), compared to that in the condenser of the open-circuit flow process (one column) as shown in Table 1.

### 3.3. Long-Term Repetitive Operation of the VOC-Reduction Process

To examine the time trend of TVOC-removal efficiency, we repeated five cycles (one cycle consisted of two condensation/adsorption processes and one desorption/condensation process) with the closed-loop flow process, which showed the better performance for VOC removal, compared to the open-circuit flow process (Figure 3). During the five cycles of operation for condensation/adsorption, the average TVOCs concentrations for each cycle were as follows: 57.5 ± 10.6 ppmv (first cycle), 53.6 ± 5.8 ppmv (second cycle), 46.7 ± 5.6 ppmv (third cycle), 49.6 ± 11.5 ppmv (fourth cycle), and 51.0 ± 12.3 ppmv (fifth cycle). The TVOC-removal efficiency was correspondingly calculated as follows:

98.5 ± 0.3% (first cycle), 98.2 ± 0.4% (second cycle), 98.6 ± 0.1% (third cycle), 98.4 ± 0.3% (fourth cycle), and 98.5 ± 0.0% (fifth cycle). The TVOC-removal efficiency was stable across the five cycles, indicating the robustness in the performance of our VOC-removal process in the repetitive operation. The time trend of TVOC-removal efficiency using the open-circuit flow process is available in Supplementary Materials (Figure S4). We found that the removal efficiency of the closed-loop flow process was higher than that of the open-circuit flow process throughout the five cycles.

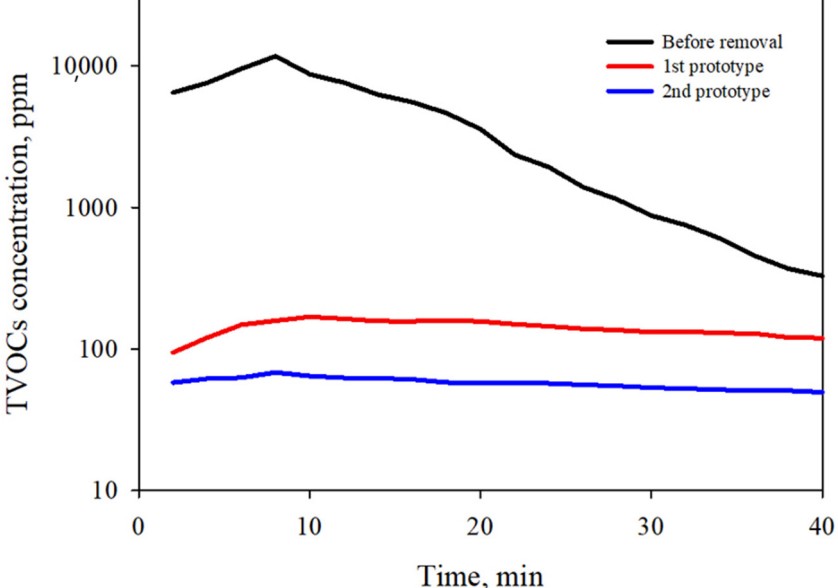

**Figure 2.** TVOC concentration in the exhaust gas before and after applying the integrated treatment processes. Black line represents TVOC concentration before VOC removal. The red line and blue line represent the TVOC concentration after applying the open-circuit flow process and the closed-loop flow process, respectively.

### 3.4. Process Simulation

Figure 4 shows the process simulation results using a pore and surface diffusion model (PSDM, AdDesignS$^{TM}$). With decane as a representative compound, the concentrations at the downstream of the adsorption process ranged between 0 (desorption stage) and 51 (adsorption stage) ppmv during 30 cycles. The concentration curves in Figure 5 did not increase or diverge across the 30 cycles, indicating that our integrated treatment process showed the most robust performance in long-term continuous operation for VOC removal. We also observed that the process simulation results for the adsorption stage (51 ppmv) were consistent with the five-cycle real measurement (46.7–57.5 ppmv) shown in Section 3.3.

### 3.5. Reduction in POCP-Weighted and SOAP-Weighted Emission

We found that 56.8 g/cycle of POCP-weighted emissions occurred after applying the VOC-reduction process, while 591.6 g/cycle occurred without the VOC-reduction process, leading to a 90.4% reduction in POCP-weighted emission (Figure 5a). The four dominant compounds in the exhaust gas before applying the VOC-reduction process were undecane (32.56%), dodecane (28.86%), decane (18.88%), and nonane (15.50%) (Figure 5b). After applying the VOC-reduction process, the four dominant compounds were decane (36.24%), nonane (35.27%), undecane (23.24%), and octane (2.98%). VOCs with higher molecular weight tend to have lower vapor pressure, resulting in higher removal efficiency in a condenser [35]. This may be the primary reason for the changes in the dominant compounds and their composition before and after VOC removal.

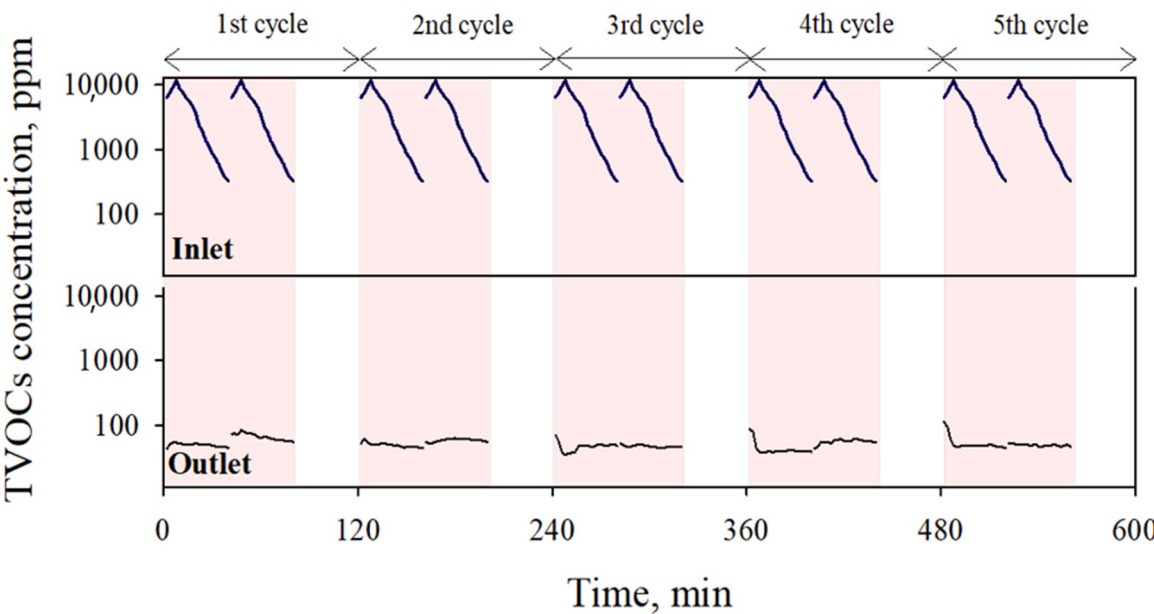

**Figure 3.** TVOCs concentrations of five cycles (one cycle consisted of two condensation/adsorption processes and one desorption/condensation process) with the closed-loop flow process applied.

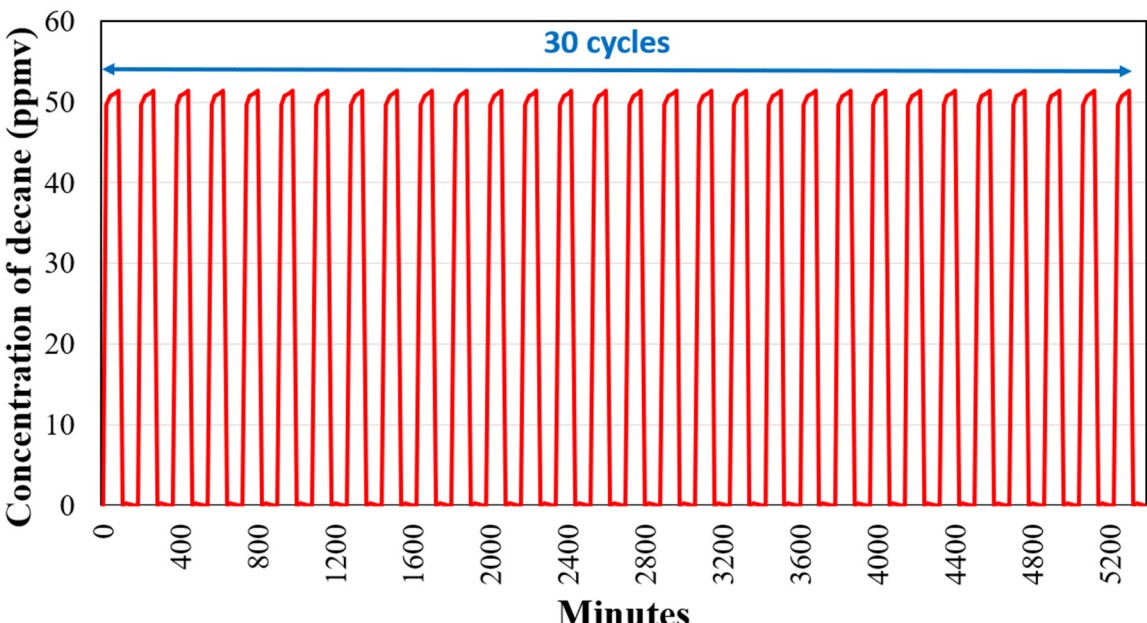

**Figure 4.** Concentration of decane (ppmv) during 30 cycles of adsorption/desorption operations. One cycle (140 min) consisted of two adsorption processes (2 × 40 min) and one desorption process (60 min).

We also found that SOAP-weight emission decreased from 250.2 g/cycle (before VOC removal) to 9.5 g/cycle (after VOC removal), leading to 95.9% reduction in SOAP-weighted emission (Figure 6a). The dominant compounds and their composition also changed before and after VOC removal. The four dominant compounds before VOC removal were dodecane (56.88%), undecane (33.90%), decane (6.75%), and nonane (1.68%) (Figure 6b). After applying the VOC-reduction process, the four dominant compounds were undecane (56.28%), decane (33.71%), nonane (8.91%), and dodecane (0.48%). Compounds with higher molecular weight, such as dodecane, undecane, and decane, had higher reductions in SOAP-weight emissions after VOC removal: dodecane: 99.9% reduction; undecane: 93.1% reduction; and decane: 79.4% reduction.

### (a) POCP-weighted emission before and after VOCs removal

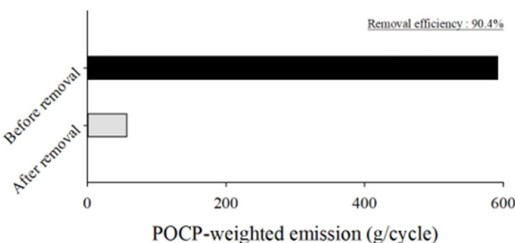

### (b) Proportion (%) of POCP-weighted emission of each compound

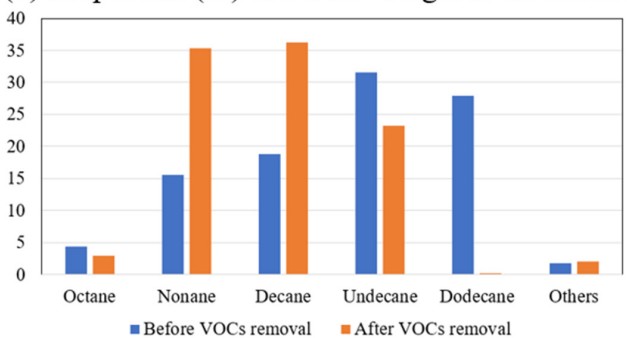

**Figure 5.** POCP-weighted emission and its proportion of each compound before and after applying the integrated treatment process.

### (a) SOAP-weighted emission before and after VOCs removal

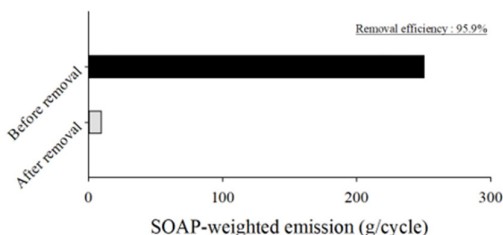

### (b) Proportion (%) of SOAP-weighted emission of each compound

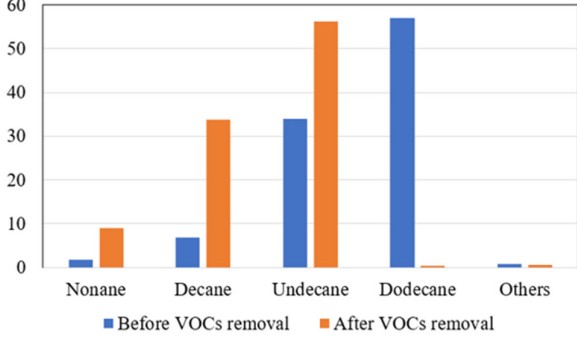

**Figure 6.** SOAP-weighted emissions and the proportion of each compound before and applying the integrated treatment process.

## 4. Conclusions

In the current study, we suggested integrated treatment processes for removing VOCs released from laundry dryers. Using the integrated treatment processes, we assessed the VOC-removal efficiency of each process in a continuous repetitive operation and obtained more than 98% VOC-removal efficiency. The average TVOC concentration in exhaust gas decreased from 4100 ppmv (before VOC removal) to 60 ppmv (after VOC removal). We also examined the robustness of our process in a continuous operation (30 cycles) using a process simulation model and observed the robust performance of our process during the continuous operation. We also quantitatively assessed the effect of our VOC-reduction process on atmospheric environment and obtained 90.4% reduction in POCP-weighted emissions and 95.9% reduction in SOAP-weighted emissions. Based on our findings in this study, we concluded that (1) our integrated treatment processes for VOC removal are applicable to small-scale laundry shops releasing high concentration VOCs intermittently, and (2) the VOC reduction using our integrated treatment processes is beneficial to the atmospheric environment with respect to ozone and SOA generation. Further studies are recommended to investigate VOC emission sources in urban areas besides small-scale laundry shops (e.g., restaurants, printing shops) and examine the applicability of our integrated VOC-reduction process.

**Supplementary Materials:** The following are available online at https://www.mdpi.com/article/10.3390/pr9091658/s1: Figure S1: Conceptual layouts of two different integrated treatment processes: (a) a process with open-circuit flow in regeneration mode and (b) a process with closed-loop flow in regeneration mode; Figure S2: Condensation efficiency of vaporized dry-cleaning solvents from the dry-cleaning process as a function of applied temperature and inlet TVOC concentration; Figure S3: The effect of process parameters on adsorption and desorption on activated alumina as an adsorbent; Figure S4: TVOC concentrations of five cycles (one cycle consisting of two condensation/adsorption processes and one desorption/condensation process) with the open-circuit flow process applied; Figure S5: Toluene concentration during an adsorption/desorption experiment in a laboratory scale; Table S1: Input parameters for adsorption/desorption model simulation.

**Author Contributions:** Conceptualization, M.S. and C.C.; methodology, M.S. and C.C.; software, M.S. and K.K.; validation, K.K. and D.K.; formal analysis, M.S.; investigation, M.S.; resources, D.K.; data curation, M.S.; writing—original draft preparation, M.S. and K.K.; writing—review and editing, K.K. and D.K.; visualization, M.S. and K.K.; supervision, D.K.; project administration, D.K.; funding acquisition, D.K. All authors have read and agreed to the published version of the manuscript.

**Funding:** This work was funded by the Ministry of Science (Sejong-si, Korea), ICT (2017M1A2A2086647).

**Institutional Review Board Statement:** Not applicable.

**Informed Consent Statement:** Not applicable.

**Data Availability Statement:** All data have been included in this manuscript.

**Acknowledgments:** This work was supported by the Technology Development Program to Solve Climate Changes of the National Research Foundation (NRF) funded by the Ministry of Science, ICT (2017M1A2A2086647). K. Kim acknowledges the financial support from Seoul National University of Science and Technology.

**Conflicts of Interest:** The authors declare no conflict of interest.

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
