# Peer review of "Reduction of Volatile Organic Compounds (VOCs) Emissions from Laundry Dry-Cleaning by an Integrated Treatment Process of Condensation and Adsorption"

_processes, doi:10.3390/pr9091658_

Round 1
Reviewer 1 Report
Volatile organic compounds (VOCs) have aroused concerns due to their health effects as well as contribution to the production of ozone and secondary organic aerosol in atmospheric environments. High levels of VOCs are intermittently emitted from small-scale laundry shops in urban areas, influencing urban atmospheric environment. In this study, the authors demonstrated efficient removal of VOCs released from small-scale laundry shops via an integrated VOCs treatment processes which incorporate condensation and adsorption in series. The work is suitable for the journal and I would recommend the acceptance of the manuscript as long as the following questions are properly answered.
- In Materials and Methods section, besides the primary ingredients of the solvents, what does the rest 5% consist of? How about their POCP and SOAP?
- In Results and Discussion section, 3.1. Determination of VOCs reduction process parameters, why toluene was chosen as the representative compound?
- What is the cost of the present treatment processes and how about the potential for applying to other scenarios besides the small-scale laundry shops.

Reviewer 2 Report
This study uses a comprehensive treatment process to help reduce the production of ozone and secondary organic aerosols. The experiment in this article is properly designed, and the experimental data obtained have certain scientific significance.
In section 2.3, please specify the analytical detection limit and the applicable range of analytical concentration.
In section 2.5, if you use the sum of individual VOCs estimations, please reconsider the applicability of the two listed equations. In addition, please attach the chromatogram of GC-MS.
In Figure 2, it is impossible to see any variation of the concentration. Can the reduction rate be used on the y-axis? Please also correct Figure 3.
